# A Review on Functionalized Graphene Sensors for Detection of Ammonia

**DOI:** 10.3390/s21041443

**Published:** 2021-02-19

**Authors:** Xiaohui Tang, Marc Debliquy, Driss Lahem, Yiyi Yan, Jean-Pierre Raskin

**Affiliations:** 1ICTEAM Institute, Université Catholique de Louvain (UCLouvain), Place du Levant, 3, 1348 Louvain-la-Neuve, Belgium; xiaohui.tang@uclouvain.be (X.T.); yiyi.yan@uclouvain.be (Y.Y.); jean-pierre.raskin@uclouvain.be (J.-P.R.); 2Materials Science Department, University of Mons, 56, Rue de l’Epargne, 7000 Mons, Belgium; 3Materia Nova ASBL, 3, Avenue N. Copernic, 7000 Mons, Belgium; Driss.Lahem@MATERIANOVA.BE

**Keywords:** chemical vapor deposition graphene, reduced graphene oxide, functionalized graphene, ammonia detection, conducting polymers, field-effect transistor sensor

## Abstract

Since the first graphene gas sensor has been reported, functionalized graphene gas sensors have already attracted a lot of research interest due to their potential for high sensitivity, great selectivity, and fast detection of various gases. In this paper, we summarize the recent development and progression of functionalized graphene sensors for ammonia (NH_3_) detection at room temperature. We review graphene gas sensors functionalized by different materials, including metallic nanoparticles, metal oxides, organic molecules, and conducting polymers. The various sensing mechanism of functionalized graphene gas sensors are explained and compared. Meanwhile, some existing challenges that may hinder the sensor mass production are discussed and several related solutions are proposed. Possible opportunities and perspective applications of the graphene NH_3_ sensors are also presented.

## 1. Introduction

Ammonia (NH_3_) is a colorless gas with a pungent smell. Most of NH_3_ in our life environment is emitted directly or indirectly by agricultural sector, waste handling, road transportation, industrial applications, and human activities. NH_3_ plays a decisive role in particulate matter (PM) formation in polluted environments [1]. NH_3_ is caustic and hazardous to human health. Specifically, it injures the human eyes, skin, respiratory tract, liver, and kidneys beyond a concentration of 25 parts per million (ppm) [2,3]. Hence, NH_3_ sensing and monitoring are essential for industrial emission control, environment conservation, and human safety. Recently, NH_3_ sensors have found new potential in medical applications, because NH_3_ is a biomarker for renal and ulcer diseases [4]. Therefore, NH_3_ sensors can be used to diagnose these diseases through detecting NH_3_ concentration from human breath. For a healthy person, the mean concentration of NH_3_ in the breath exhalation is about 0.83 ppm [5]. The patients with renal disorders or ulcers exhale NH_3_ concentration in the range from 0.82 to 14.7 ppm (the mean value is 4.88 ppm) [6]. Although many NH_3_ sensors have been investigated in the literature [7,8,9], it is still a great challenge to develop cost effective, low power consumption, working at room temperature, and compact sensors with high sensitivity, great selectivity, rapid response, and good recovery. Moreover, it is urgent to develop NH_3_ intelligent sensors, which can be integrated with complementary metal-oxide semiconductor (CMOS) circuits and used in the context of the Internet of Things (IoT). Metal oxides, such as SnO_2_, V_2_O_2_, WO_3_, ZnO, In_2_O_3_, TiO_2_, and Cu_2_O, are very popular compounds for NH_3_ sensing [10,11,12]. However, even though they possess a very low detection limit, they exhibit poor gas selectivity. The other flaw of metal-oxide-based sensors is high-temperature operation, which brings about large power consumption, safety hazard, short lifetime, and high cost. Other classical materials for NH_3_ sensing are based on conducting polymers such as polypyrrole or polyaniline [13,14,15,16,17]. Different authors proposed hybrid structures mixing conducting polymers with metal oxides or even carbon nanotubes [18,19,20]. Though those sensors can work at room temperature, they are also sensitive to humidity. Moreover, their lifetime is limited.

Since the discovery of graphene (G) in 2004, it has been researched and developed as a gas-sensing material because of its extremely large surface-to-volume ratio and high carrier mobility. One adsorbed molecule can cause a notable change in the graphene resistance, allowing the sensing ability to single molecule level [21,22]. In 2007, Schedin et al. [23] first demonstrated the detection of a single NO_2_ molecule by using pristine graphene in pure helium or nitrogen at atmospheric pressure. Their results also revealed that pristine graphene sensor responds not only to NO_2_, but also to NH_3_, CO, and H_2_O. Subsequently, graphene-based sensors were widely exploited for detecting various types of gases, for example NH_3_, CO, O_2_, NO_2_, H_2_, CH_4_, SO_2_, H_2_S, and volatile organic compounds (VOCs) [24,25,26,27]. Yavari et al. [28] used chemical vapor deposition graphene (CVD-G) sensor to simultaneously detect the trace amounts of NO_2_ and NH_3_. The detection limit reached 100 and 500 parts-per-billion (ppb) for NO_2_ and NH_3_, respectively. These values are markedly superior to commercially available sensors. However, the full recovery of pristine graphene sensors requires them to be heated at 200 °C in inert gases. Dan et al. [29] found out that clean graphene did not response to NH_3_ at room temperature. The first-principles calculation also predicts that pristine graphene is chemically inert [30]. These facts clearly indicate that pristine graphene is nonselective and weakly binding with NH_3_ molecules in ambient condition. As a result, interest in the functionalization to generate specific links between pristine graphene and NH_3_ molecules is highlighted. Generally, graphene functionalization can be accomplished by metallic nanoparticles, metal oxides, organic molecules, or conducting polymers. This article does not review all the functionalization methods presented in the literature but mainly focuses on the graphene functionalized by ultrathin conducting polymers for NH_3_ sensing. 

The article is arranged as follows. In Section 1, we describe the electronic structure of graphene, summarize graphene physical properties suitable to sensing application, and present graphene preparation methods. In Section 2, we briefly review the operation modes and related working principles of various functionalized graphene sensors for NH_3_ detection, including mass sensitive, Schottky/semiconductor diode, chemiresistor, and field-effect transistor (FET) sensors. In Section 3, the fabrication methods of various graphene NH_3_ sensors are first briefly described, followed by the presentation of their main properties and their related sensing mechanisms. The graphene sensors functionalized by ultrathin polymers are presented in detail. In Section 4, some existing challenges that may hinder the sensor mass production are analyzed, and several related solutions are proposed. Finally, in Section 5, possible opportunities and perspective applications of the graphene NH_3_ sensors are presented, which may provide future research directions. 

## 2. Characteristics and Preparation of Graphene

Graphene is one of the most promising materials for gas-sensor applications. In this section, we present the distinctive band structure and outstanding physical properties of graphene. Then, we briefly review the graphene preparation methods.

### 2.1. Energy Band Structure of Graphene

Graphene is a two-dimensional (2D) material with a monolayer of *sp*^2^-bonded carbon atoms. Electrons are delocalized and free to move in this 2D sheet. The energy band structure of graphene is illustrated in Figure 1 [31]. The valence and conduction bands of graphene are conical valleys, which are attached at Dirac points. The energy dispersion curve is linear around the Dirac points, leading to an extremely large mobility of the charge carriers. Pristine graphene is a zero band-gap semiconductor, and its Fermi level is located at the Dirac point, corresponding to zero conductivity at T = 0 K under vacuum [32]. In practical situations, graphene possesses defects that cause a *p*-type doping [33]. The zero or quasi-zero bandgap of pristine graphene can be a hurdle to use it as a sensing layer [34]. However, the electron transfer with target gas molecules leads to the Fermi level shift. More precisely, when graphene reacts with electron-donor or electron-acceptor gas molecules, the zero conductivity disappears. Moreover, doping or defects can also shift the Fermi level, which makes graphene become a *p*-type or *n*-type semiconductor.

Due to this structure, graphene is very sensitive to Fermi level changes. Its conductivity can be easily modified by doping or a back gate voltage when the graphene sheet is fabricated as a field effect transistor sensor [35].

### 2.2. Physical Properties of Graphene

Single-layer graphene has the largest surface-to-volume ratio of 2630 m^2^/g [36] and exposes all carbon atoms to environment. This can provide the largest number of binding sites per unit volume to yield high sensitivity for sensors. The high carrier mobility (200,000 cm^2^/s.V) [37] and excellent electrical conductivity (1738 Siemens/m) [38] of graphene at room temperature inherently ensure low electrical noise and energy consumption in graphene sensors. Carrier concentration in graphene strongly affects the graphene resistance, and it can be modulated by the adsorption of target molecules injecting electrons or holes into graphene. The significant change, coupled with a low electrical noise, of the graphene resistance makes it have the ability for detecting a single molecule. Particularly, graphene is easy to be functionalized and compatible with organic molecules through a π–π stacking interaction and/or electrostatic interaction [39]. For graphene oxide (GO), the oxygen-containing functional groups (such as hydroxyl and epoxy) are attached to honeycomb-like carbon of graphene. Therefore, GO can be readily modified with target molecules through the functional groups. These facts make graphene and graphene oxide ideal materials for designing gas sensors.

### 2.3. Preparation Methods of Graphene

Historical graphene was first prepared by mechanical exfoliation of graphite [35,40]. For example, highly oriented pyrolytic graphite (HOPG) is attached to the photoresist layer over a glass substrate. Using scotch tape, thick flakes of graphite are repeatedly peeled off. Thin flakes left in the photoresist are released in acetone. The target substrate (such as SiO_2_/Si wafer) is dipped in the acetone and then washed with plenty of water and propanol, which removes mostly thick flakes. In this case, some thin flakes are captured on the target substrate’s surface due to van der Waals and/or capillary forces. Although mechanical exfoliation can provide high-quality single-layer graphene, it is not a scalable technique, and other techniques were proposed.

Chemical vapor deposition (CVD) holds great promises for large-scale production of high-quality graphene [41,42]. Conceptually, graphene CVD is a simple technique: it involves the decomposition of hydrocarbon precursors (such as CH_4_) on catalytic metal substrates at high temperature (about 1000 °C) in a controlled atmosphere or atmospheric pressure. Cu, Ni, Ti, Ru, Pd, Pt, Ir, Co, Au, and Rh can be used as the catalytic metal substrates [43]. Among them, Cu is a very attractive catalyst and extensively chosen, since very low carbon solubility of Cu allows self-limited graphene growth, leading to highly homogeneous graphene sheets.

Graphene has also been grown on the surface of single-crystal silicon carbide (SiC) by thermal decomposition [44]. Namely, Si sublimation from SiC surface yields single-layer or multi-layer graphene structure at the graphene–SiC interface. However, it is quite difficult to exfoliate or transfer from that the SiC substrate.

For large-scale production of low-cost graphene, one of the most popular approaches is the use of strong oxidizing agents to obtain GO and then convert GO to reduced graphene oxide (rGO) by thermal treatment or chemical reduction. In 1958, Hummers [45] reported a method for the GO synthesis by using KMnO_4_ and NaNO_3_ in concentrated H_2_SO_4_. This method has broadly been used in gas-sensor fabrication because of its high efficiency. However, it still has a few drawbacks. For instance, the oxidation procedure releases toxic gasses and the control of the graphene oxide sheets is difficult. Recently, several modified methods were proposed in the literature to improve the control of the GO and rGO synthesis [46,47]. It is worth noting that the reduction in GO is not complete, and there are defects and OH groups in the final rGO [48,49].

In brief, the outstanding properties of graphene make it suitable for developing highly sensitive sensors for NH_3_. Graphene and rGO can be fabricated by various methods. CVD can provide large-scale production of high-quality graphene. However, high power consumption and waste catalytic metals limit its production cost. The rGO is considered as a more practical method due to its large-scale and low-cost production.

## 3. Working Principle of Various Graphene-Based NH_3_ Sensors

In this section, we briefly review the working principle and operation mode of various functionalized graphene sensors for NH_3_ detection. It includes mass sensitive sensors, Schottky/heterojunction diode sensors, chemiresistor sensors, and field-effect transistor (FET) sensors.

### 3.1. Mass Sensitive Sensor

Mass sensitive sensor is a transducing device with a sensitive layer. After the sensitive layer adsorbing the target gas, the device gives a signal related to the mass uptake of this layer. The sensors are divided in quartz crystal microbalance (QCM) sensors [50] and surface acoustic wave (SAW) sensors [51]. The working principle of the QCM sensor is based on the change in mass or thickness of the sensitive layer adhering to the surface of a quartz crystal. Specifically, the resonance frequency of the QCM sensor changes with the concentration of the target gas adsorbed on the sensitive layer. As shown in Figure 2, the QCM piezoelectric system is composed of a quartz crystal resonator disc with electrodes deposited on each side. When NH_3_ molecules adsorb on the functionalized graphene surface over the quartz crystal surface, the resonant frequency of the resonator varies proportionally with the mass of NH_3_ adsorbed on the sensing layer. The change in the resonance frequency is used to measure the mass of adsorbed NH_3_ by using the Sauerbrey equation [52]:(1)$$\Delta f=-2.26\times 10^{-2}f_{0}{}^{2}\times \frac{\Delta m}{A}$$
where Δf and Δm are the change in the resonance frequency and the addition of mass on the QCM after exposing to NH_3_, respectively. $f_{0}$ is the fundamental resonance frequency of the QCM, and *A* is the surface area of the electrode. According to the chemical equilibrium, one can obtain the Δm direct relation with the concentration in the gas phase. The use of polymer-functionalized graphene enhances the NH_3_ adsorption and increases the resonance frequency change, thereby improving the sensitivity of the QCM.

The surface acoustic wave (SAW) sensor is more common than the QCM sensor. Its working principle is based on the elastic loading effect. Figure 3 shows the schematic diagram of the SAW sensor, which has an input interdigitated transducer (IDT) and an output IDT built on both sides of a piezoelectric substrate [53]. The space between the input IDT and the output IDT is called the delay line. When an alternative voltage (AC) is applied to the input IDT, a surface acoustic wave (deformation of the surface of the piezoelectric substrate) is generated and propagates to the output IDT. Once the SAW reaches the electrodes of the output IDT, the deformation of the piezoelectric substrate is transformed into an electrical signal (voltage wave). The SAW travels much slower (speed of sound in the piezoelectric substrate) compared to an electromagnetic wave, resulting in an appreciable delay. If a coating is deposited in the space between the IDTs, the propagation of the surface acoustic wave is modified, which can be measured by a change in the delay time and/or the intensity of the output voltage. The adsorption of molecules on this coating (change of mass) will modify the propagation of the surface acoustic wave leading to a measurable change in the delay time or attenuation, which is exploited for manufacturing mass sensitive sensors. Interestingly, the electrical signal change is proportional to the gas concentration. With such a SAW sensor using a spin coated rGO layer, Tang et al. [54] obtained a detection limit of 500 ppb for NH_3_ detection. This is attributed to the abundant oxygen-containing groups on the surface of rGO film, which strongly adsorb NH_3_ molecules and increase the film stiffness.

### 3.2. Graphene/Semiconductor Schottky Diode Sensor

Graphene (as the metal electrode) and semiconductor Schottky diode sensor is a promising candidate for quickly sensing low concentration gas. When the Schottky diode is exposed to the target gas, a small change in the graphene electrode composition and Schottky barrier height (*ψ*_SBH_) can lead to a huge difference in the diode current–voltage (*I*–*V*) characteristics. Figure 4 shows the schematic energy band diagram of a graphene/*n*-type silicon interface for the pristine graphene state (middle), the graphene electrode exposed to electron-donor gas (left), and the graphene electrode exposed to electron-acceptor gas (right) [55]. NH_3_ molecule is strongly nucleophilic. Once it is attached to graphene, it donates electrons to the latter via coupling π bonds. One NH_3_ molecule can donate 0.03 electrons onto graphene [56,57]. The working principle of the Schottky diode for NH_3_ can be described as follows: when the graphene electrode surface is exposed to NH_3_, the electron transfer from NH_3_ to graphene causes a shift of the graphene Fermi level toward the Si conduction band, lowering the *ψ*_SBH_ (as shown in Figure 4 left). The current (*I*) across the graphene/*n*-type Si interface is determined by the *ψ*_SBH_ through formula [58]:(2)$$I=\;AA**T^{2}\exp(-q\frac{\psi_{\mathrm{SBH}}}{kT})[\exp(\frac{qV}{nkT})\;-\;1]$$
here, *V* is the applied voltage, *A* is the area of the Schottky junction, *A*** is the Richardson constant, *T* is temperature, *q* is electric charge, *k* is the Planck constant, *ψ*_SBH_ is the Schottky barrier height, *n* is the ideality factor. For the pristine graphene/*n*-type silicon, *n* = 1.41 and *ψ*_SBH_ = 0.79 eV. For the pristine graphene/*p*-type silicon, *n* = 1.31 and *ψ*_SBH_ = 0.74 B eV [55].

The chemical or physical modification of the graphene electrode by NH_3_ directly impacts the current of the Schottky diode. More specifically, a small change in the *ψ*_SBH_ caused by NH_3_ adsorption on the graphene electrode can exponentially change the current of the Schottky diode through Equation (2). As a consequence, the current change can be used to identify various adsorbents and determine their concentration. Figure 5 shows the schematic *I*–*V* curves of a graphene/*n*-type Si Schottky diode sensor before and after exposing to electron-donor gas and electron-acceptor gas.

Polichetti et al. [59] reported a rGO/*n*-Si Schottky diode for sensing NH_3_. Their results indicated that 200 ppm NH_3_ caused the *ψ*_SBH_ change of 0.1 meV. In order to engender selectivity of the graphene/Si Schottky diode sensor, functionalizing graphene is necessary. Biswas et al. [60] used rGO sheets functionalized by polyaniline (PANI) as the metal electrode of the Shottky diode sensor to selectively detect NH_3_ in ambient air. They obtained a detection limit of ppb level with the response and recovery times of 95 and 101 s, respectively.

### 3.3. Graphene Chemiresistor Sensor

Chemiresistor sensor is popular for the successful detection of a wide range of gases. Chemiresistor sensor is easy to fabricate, has relatively low cost, and enables direct measurement. The typical structure of the graphene chemiresistor sensor is shown in Figure 6. It consists of interdigitated electrodes (IDEs) on a substrate covered with graphene or various graphene composite materials. The resistance of the sensor is altered by the reversible electron transfer between the adsorbed target gas and the sensing layer. The substrate is usually made of a thin silicon dioxide (SiO_2_) layer on top of Si wafer. The SiO_2_ thickness is 90 or 300 nm for easily observing graphene with an optical microscope [61]. The use of the IDEs is more convenient to contact small pieces of graphene, such as rGO flakes. For large size of graphene, such as CVD-G, two electrode pads without IDEs can be used.

In ambient condition, graphene behaves like a *p*-type semiconductor containing many holes [62], as shown in Figure 7a. Its Dirac point shifts to the right in Figure 7b. When electron-donor gas molecules (such as NH_3_) are adsorbed on the graphene surface, they donate the weakly bonded electrons to graphene, causing the Fermi level to rise by small increments. In this case, the donor electrons deplete the holes in the *p*-type graphene, resulting in the resistance increase in graphene. Conversely, the electron-acceptor gas molecules (such as NO_2_) are adsorbed on the graphene surface, they take the electrons from graphene, lowering the Fermi level. The extra holes increase the holes concentration in the *p*-type graphene, leading to the resistance decrease in graphene. Therefore, the gas concentration can be quantitatively analyzed by directly measuring the change in the sensor resistance under the interaction with the target gas [63]. The case of the *n*-type graphene can be described by only changing sign referred to in Figure 7 (the left).

The electron-donating or electron-accepting nature of the gas is related to the relative position between the graphene Fermi level and the electron energy level of the highest occupied molecular orbital (HOMO) or the electron energy level of the lowest unoccupied molecular orbital (LUMO).

### 3.4. Graphene FET Sensor

Many research groups used graphene to fabricate field-effect transistors (FET) [64]. However, its quasi-zero bandgap and low on/off-current ratio hinder its potential as a basic unit for digital and analog electronics. On contrary, the bipolar band structure of graphene makes it extremely suitable for FET sensors, because the charge carrier density in graphene can be easily and continuously tuned by the gate voltage [65,66]. Compared with the most common chemiresistor sensors, the graphene FET sensor is regarded as high sensitivity and great selectivity due to the ability of delivering multi-parameter response, namely, drain current, threshold voltage, flat band voltage, work function, and on/off-current ratio of the sensor.

The detection mechanism of the graphene FET sensor to gas depends on the synergistic effect: the current of the sensing layer is modified (i) by the field strength through the gate voltage and (ii) the electron transfer with the gas molecules. Figure 8 shows the schematic cross-section of the graphene FET sensor, which is composed of a source, a drain, a graphene channel (acting as the sensing layer), and a back gate. The source is grounded, and the drain has the highest applied potential. The sensing layer between the source and the drain contacts lies on a thin insulator. The highly-doped Si substrate can act as a back gate electrode. There are two types of sensing layers: *p*-type and *n*-type, for which the current flow is due to electron and hole transport, respectively. Hereafter, we will limit our analysis and discussion to the *p*-type sensing layer, since the principle is the same for *n*-type sensing layer but changing sign. When the target gas is off, the current into the sensing layer is only controlled by the back gate voltage (*V*_G_). As shown in Figure 8a,b, positive gate voltage raises electron conduction whereas negative gate voltage promotes hole conduction. As shown in Figure 8c, when the sensing layer is exposed to the electron-donor gas under positive gate voltage, the channel electrons induced by positive gate voltage and the donated electrons by target gas, strongly increase the sensing layer current. In other words, the amplitude of the sensing current signal is amplified, equivalent to increasing sensitivity. Figure 8d indicates that when the *p*-type sensing layer is exposed to the electron-donor gas under the negative gate voltage, three possibilities exist: (i) the channel holes induced by negative gate voltage are equal to the donated electrons by the target gas, the sensing layer current keeps unchanged (keeps on baseline); (ii) the induced channel holes are more than the donated electrons, the sensing layer current decreases; (iii) the induced channel holes are less than the donated electrons, the sensing layer current increases. In the case of the *p*-type graphene FET sensor with electron-donor gas, it is better to use high positive gate voltage for the measurement. Such choice ensures that the graphene FET sensor would still be with electron conduction under electron-donor exposure. Otherwise, the case will be very complicated. It was reported [67] that for detecting NH_3_, the *p*-type graphene FET sensor exhibits superior performance operated at high positive gate voltage rather than at negative gate voltage. On the contrary, for the electron-acceptor gas, it is better to use *n*-type graphene and high negative gate voltage.

When the target gas is off, the simplified drain current (*I*_D_) of the graphene FET sensor is expressed by [68]
*I*_D_ = *µW/L C*_OX_ (*V*_G_ − *V*_D_/2)*V*_D_(3)
where *µ* is the carrier mobility, *W* and *L* are the width and length of the sensing layer, respectively, *C*_OX_ is the unity gate capacitance, *V*_D_ and *V*_G_ are the drain voltage and the gate voltage, respectively. When the sensing layer is exposed to the target gas and the modification of the graphene surface is taken into account in the above relationship, the following equation is obtained [69]:
*I*_D_ = *µW/L*[*C*_OX_(*V*_G_ − *V*_D_/2) + *Q*_m_]*V*_D_(4)here, *Q*_m_ (C.cm^−2^) is the interface charges induced by the gas molecules. Importantly, *Q*_m_ is not zero when electrons transfer between the gas molecules and graphene channel. This is the reason why the graphene FET sensor can detect the gas concentration when it reacts with an electron-donor or electron-acceptor gas. From Equation (4), we can clearly see the synergistic effect: for a given sensing layer and drain voltage (*V*_D_), *I*_D_ depends on not only the gate voltage (*V*_G_), but also the adsorbed gas molecules (*Q*_m_).

Gautam et al. [70] used a back-gated graphene FET for NH_3_ sensing. Their sensor was fabricated by CVD-G and the sensitivity reached the ppm level. They also measured the relationship between the Dirac point shift and the NH_3_ concentration at different temperatures. When the sensor was tested from room temperature to 100 °C, the Dirac point shift was found to increase linearly with temperature and the concentration of NH_3_. The average shift of Dirac point of 0.33 V/ppm was obtained at room temperature.

As discussed above, graphene-based materials are suitable for various mode NH_3_ sensors. Their main characteristics are summarized as follows. The mass sensitive sensors have the detection limit down to the ppb level, but they need additional apparatus for indirect measurements. The graphene/semiconductor Schottky diode is a promising mode for rapidly sensing a low concentration of gas. However, the influence of humidity and interference gas is an issue. Although the chemiresistor sensor is the simplest mode, low signal/noise ratio limits its sensitivity. The graphene FET sensor is regarded as a good candidate owing to the ability of delivering multi-parameter response. However, the measurement technique should be improved, and the power consumption should be reduced. For example, an alternating current (AC) is applied to the gate of the graphene FET sensor in order to simultaneously excite the perturbation of target molecular dipoles and modulate the channel charge carriers. 

## 4. Functionalized Graphene NH_3_ Sensors

Pristine graphene without bandgap behaves like a semimetal [71]. It is not a good choice for the NH_3_ detection, because it does not possess any functional groups or defects. Monocrystalline graphene of mechanical exfoliation is well known as pristine graphene. However, it includes few isolated point defects. When NH_3_ molecules are attached on the isolated point defects, the resistance of the pristine graphene sensor does not significantly change. Since the donor electrons of NH_3_ can transfer through low resistance pathways around the isolated point defects [72]. On the contrary, functionalized graphene is one of the most promising materials for the NH_3_ sensor. Thanks to the abundant defects (grain boundaries, line defects, or a few point defects) and functional groups on it, they strongly adsorb NH_3_ molecules. Around such defects, no low-resistance pathways exist. So, the resistance change caused by NH_3_ adsorption is significant, leading to a higher sensitivity.

It is mentioned above that non-functionalized graphene is nonselective to gases [73]. A variety of gas molecules can be adsorbed on it to give similar resistance changes. For *p*-type graphene, the adsorption of a reducing gas (either NH_3_ or H_2_S) donates electrons to the graphene and depletes the concentration of holes, increasing the graphene resistance. The adsorption of an oxidizing gas (either NO_2_ or SO_2_) accepts electrons from the graphene and enhances the concentration of holes, decreasing the graphene resistance. As a consequence, different gases may generate the same sensing signals [74]. Indeed, the first non-functionalized graphene sensor can adsorb not only NH_3_, but also CO, H_2_O, and NO_2_ in the same conditions as shown in Figure 9 [23], proving its poor selectivity. Ultraviolet or infrared radiation is able to make graphene sensitive to different types of gases, allowing for certain selectivity. However, the light adsorption efficiency of graphene is weak [75,76]. The selective capabilities of graphene can be easily enhanced by grafting functional groups on its surface. This strategy is followed by most of the researchers.

The functionalization (generating specific links and increasing the adsorbing sites) is a viable method to improve chemical inert and poor selectivity of graphene. The graphene functionalization can be classified to covalent and non-covalent strategies [77]. The former transforms graphene’s π-orbitals from *sp*^2^ into *sp*^3^, disrupting graphene electronic and mechanical properties. Whereas, the latter provides functional groups to the graphene surface by electrostatic or π–π interactions, enhancing the graphene bonding ability and simultaneously preserving the graphene original properties (the high carrier mobility and favorable noise characteristics). It is, thus, important to optimize and improve non-covalent functionalization techniques.

Over the past few years, a number of functionalization procedures have been developed. Various materials, from metallic nanoparticles and metal oxides to organic molecules and conducting polymers, are used to accomplish the graphene functionalization for promoting a high sensitivity and great selectivity to NH_3_ sensing. According to the interaction nature between the target molecule and the receptor attached to graphene (or rGO), sensing reactions are distinguished to physisorption and chemisorption. The physisorption depends on the van der Waals forces of attraction with low binding energy, leading to full and fast recovery, but low sensitivity and poor selectivity. The chemisorption depends on the formation of chemical bonds between NH_3_ molecules and the sensing layers, resulting in good selectivity, but slow and incomplete recovery. For certain sensing layers, physisorption and chemisorption can happen at the same time. Both adsorptions can cause charge transfer, yielding *p*-doping or *n*-doping of the graphene film.

### 4.1. Metallic Nanoparticles

One of the strategies is to modify graphene by decorating its surface with metallic nanoparticles (NPs). They were widely used for detecting hydrogen (H_2_) [78,79,80]. Recently, we developed a new chemical technique to deposit Pd nanoparticles of uniform and high-density distribution on monolayer CVD graphene for fabricating the room-temperature operation H_2_ sensors with high performance [81].

Cui et al. [82] reported an rGO chemiresistive sensor functionalized with Ag nanoparticles. The sensor had a response of 17.4% for 10,000 ppm NH_3_ at room temperature. While the rGO sensor without Ag nanoparticles only shown a response of 0.2%. Karaduman et al. [83] fabricated NH_3_ sensors based on rGO decorated by metallic nanoparticles (Ag, Au, and Pt) by using a single-step chemical reduction. They obtained the highest sensitivity of 6.52%/ppm with Ag-NPs/rGO sensor. This sensor exhibited fast response/recovery of 5 s and great selectivity compared to CO, CO_2_, and H_2_ at room temperature. Gautam et al. [84] and Song et al. [85] also presented the CVD-G sensors functionalized by Ag and Au, respectively, for NH_3_ detection. The other functionalization methods of metallic nanoparticles can be found in reference [86].

Now, we take the Pt-decorated graphene sensor as an example to explain the NH_3_ sensing mechanism. When the Pt nanoparticles are weakly bonded with graphene by van der Waals force, the electronic structure of graphene is still preserved [87]. As shown in Figure 10a, the work function of Pt and *p*-type graphene is 5.1 [88] and 4.7 eV [89], respectively. It is worth reminding that the CVD-G transferred on SiO_2_/Si substrate behaves like a *p*-type semiconductor. After the Pt nanoparticle decoration, the graphene resistance is decreased, since the work function of Pt is larger than that of the *p*-type graphene. In the other words, the Fermi level of Pt is lower than that of the *p*-type graphene. This is beneficial towards the electron transfer from the *p*-type graphene to Pt nanoparticles, enhancing the hole density in the *p*-type graphene. When the Pt-decorated graphene sensor is exposed to NH_3_ at room temperature, NH_3_ molecules are dissociated at the surface of the Pt nanoparticles to form the Pt-NH_3_ phase, which exhibits a work function smaller than the Pt one, as shown in Figure 10b. In this case, the graphene resistance is increased due to the electron transfer from the Pt-NH_3_ phase to the graphene, reducing the hole density in the *p*-type graphene.

In fact, the Pt nanoparticles act as catalysts to concentrate the reactants by adsorption to increase the probability of interaction, thereby enhancing the sensor sensitivity. Moreover, the Pt nanoparticles may also provide number of specific reaction sites for NH_3_ chemisorption, improving the sensor selectivity. The Pt-decorated graphene surface can be considered as a graphene film with *p*-type doping. It is equivalent to the hole reservoir and pathway to accelerate the sensor response and recovery. The issue for the functionalization by noble metal NPs, including Pt, Au, Ag, Pd, Ti, and so on [90,91,92], is that they are not compatible with CMOS technology due to a contamination problem.

### 4.2. Metal Oxides

In general, metal-oxide-sensing materials have relatively poor selectivity and need a high operating temperature. To overcome these shortages, heating or illumination is required. This results in a complex fabrication process, high power consumption, high cost, and the size increase in the overall device. It is interesting that the hybrid of metal oxides with graphene can significantly improve the sensing performance, particularly, the selectivity and response/recovery times at room temperature [93,94].

At room temperature, a pure SnO_2_ sensor presented a negligible response even for 100 ppm NH_3_ [95]. However, the graphene–SnO_2_ hybrid sensor yielded a response of 21% for 50 ppm NH_3_, with response/recovery times of 15/30 s and without the interference of VOCs [96]. The hybrid of the graphene–SnO sensing layer included two steps: CVD graphene was oxidized at 250 °C to create C-O bonds, and then, the oxidized graphene reacted with a SnCl_2_ solution at 70 °C to form C-O-Sn- bonds. When the hybrid sensor was exposed to NH_3_, the donor electrons of NH_3_ transferred from Sn to C of graphene through O. High sensitivity of the hybrid sensor is attributed to high electrical conductivity of graphene. Sn is a transition element, and it can easily accept the electron into its “d” orbital. Therefore, the rapid response was obtained in the hybrid sensor during exposure to NH_3_, while the rapid recovery is due to the weak bonding between NH_3_ and Sn. Kodu et al. [97] compared two types of graphene NH_3_ sensors functionalized by V_2_O_5_. The response of the epitaxial graphene sensor was one order of magnitude higher than the one of the CVD-G sensor. The authors believed that the adsorption sites provided by V_2_O_5_ nanophase were similar for both, but the response difference was due to the larger initial free charge carrier in the CVD-G sensor.

More work on graphene mixed by metal oxides can be found in our previous work [98], which summarized the latest progress of metal oxide/graphene sensors for the detection of NH_3_ at room temperature. Meanwhile, our previous review raised the improved experimental schemes and the future research directions. The functionalization methods of metal oxides are also described in the review paper [99].

### 4.3. Organic Molecules

One of the advantages of graphene is its compatibility with organic molecules [100]. Graphene can be coupled or stacked with organic molecules, such as dye molecules. Midya and Kumar [101,102] reported a new technique to combine rGO with Rose Bengal (one kind of organic dye molecules) for the selective determination of NH_3_. The combination included a lot of functional groups for selectively binding NH_3_ molecules. Therefore, the Rose Bengal/rGO sensor exhibited an enhanced response to NH_3_ compared with the rGO sensor without Rose Bengal. The detection limit of the Rose Bengal/rGO sensor was 0.9 ppm.

Duy et al. [103] demonstrated a flexible, transparent, and wearable sensor based on rGO coupled with bromophenol blue (BPB) aromatic molecules for the sensitive and rapid detection of NH_3_ in ambient condition. Their sensor has dual-mode (visual and electrical) response to NH_3_ and high stability under mechanical bending. Their results reveal that BPB aromatic molecules can be anchored and concentrated at the edges and defects of the rGO layer. Interactions between the sensing layer and NH_3_ molecules can be increased on rough surfaces. This is why the BPB/rGO sensor has an increased sensitivity. On the other hand, the BPB/rGO sensor is similar to pH papers, which can qualitatively measure the presence of NH_3_. Specifically, when the sensor was exposed to NH_3_, the color change was clearly observed with naked eyes.

Besides, porphyrins and phthalocyanines with different coordinated metals were used to functionalize graphene for improving the performance of the NH_3_ sensors [104]. For example, the Co-porphyrin/CVD-G sensor showed six times greater response to NH_3_ compared to non-functionalized graphene [105]. The Co-phthalocyanine/rGO sensor also showed a higher and faster response for low concentration of NH_3_ (~2.5% and 45 s for 100 ppb NH_3_) [106]. In addition, the Cu-phthalocyanine/rGO sensor has good selectivity to NH_3_ (to NH_3_ is 15 times higher than to CO_2_, CH_4_, H_2_, and CO) [107].

However, most of dye molecules are toxic. In the case of high concentration, Rose Bengal is toxic on the human corneal epithelium [108], and bromophenol blue is toxic on human pigmented epithelial cells [109].

### 4.4. Conducting Polymers

The selectivity of gas sensors can be achieved with conducting polymer/graphene composites at room temperature. The synergistic effect of the conducting polymer/graphene is highly significant towards enhancing the sensor performance. Conducting polymers include polypyrrole (PPy), polyaniline (PANI), polythiophene (PTh), poly (butyl acrylate) (PBuA), and poly (vinylidene fluoride) (PVDF) [110]. Hereafter, we mainly review the graphene sensors functionalized by PPy and PANI, which are frequently used due to their excellent performance, i.e., inherent flexibility, low preparation cost, simple deposition process, environmental stability, operation at room temperature, and easy compatibility with other technologies (such as CMOS) [111,112,113,114,115]. Presently, the conducting polymers have become a hot spot in the research of gas sensors.

In fact, pure conducting polymer gas sensors have attracted great interest due to the high sensitivity, repeatable response, and low-cost production [116]. The first NH_3_ sensor with PPy as sensing material goes back to 1983 by Nylander et al. [117]. To date, pure conducting polymer sensors have been able to detect a wide range of gases. Besides NH_3_ [118], they can detect NO_2_ [119], H_2_S [120], amines [121], polar vapors [122], and so on. Although Chen et al. [123] demonstrated a single PANI nanofiber FET, which presented a sensitivity of 7%/ppm NH_3_, the high contact resist (Schottky contact) between PANI and electrode materials was an issue. General problems in the use of conducting polymers for gas detection are: (i) sensing response and recovery are usually slow [124], (ii) the sensors require high operation voltages (high power consumption) to reach high sensitivity, and (iii) poor thermal stability and immunity to humidity [125].

In 2009, Dan et al. [29] first found out that a layer of polymer photoresist (about 1 nm) inevitably left on the graphene surface during photolithography process, and that the residual polymer has a significant impact on gas sensitivity. Specifically, the graphene sensor with polymer residues, such as polymethylmethacrylate (PMMA), has a strong electrical response compared with the clean graphene sensor. Graphene is chemically doped (i.e., functionalized) by these polymer residues, which can act as a gas concentrator to improve the sensing performance. A new avenue of research in graphene-based gas sensing was opened up by this finding. Consequently, the association of graphene and polymers was considered by many research groups. A practical NH_3_ sensor was built in 2012 based on PANI/graphene hybrids with a response of 59.2% for 50 ppm NH_3_ [126].

The easily tunable electrical properties of PPy make it a very popular conducting polymer [127,128]. PPy can be mainly synthesized by chemical and electrochemical methods. The other related synthesis methods were reported in reference [129]. The advantages of the chemical production method of PPy are its versatility and yield, while it features a poor reproducibility. On the other hand, with the electrochemical method, the applied current or potential enables the fine control of the preparation process. The issue of this approach is that the synthesis of PPy thin films needs a conducting substrate [130]. Therefore, graphene is a good candidate as conducting substrate.

Most of PPy/graphene sensors used the chemical synthesis PPy. Tiwari et al. [131] prepared a chemiresistive PPy/rGO sensor for the NH_3_ detection by in situ oxidative polymerization. The sensor exhibited a response three times larger than the pure PPy thin film sensor. Wang et al. [132] developed a PPy/rGO sensor, in which the assembled GO sheets over Au-IDEs were chemically reduced by pyrrole vapor. Their sensing response for NH_3_ is nearly consistent with Tiwari’s group research result. Hu et al. [133] reported a chemiresistive PPy/rGO sensor based on GO chemically reduced by pyrrole. The sensor achieved better sensing performance compared with the PPy nanofibers alone or the pure rGO sensor. The good sensing properties are attributed to rGO pristine properties as well as adsorbed PPy molecules.

According to the literature survey, we summarized in the Table 1 the currently published NH_3_ sensors based on the composites of CVD-G or rGO with conducting polymers, operating at room temperature. As we can see from the Table 1, all of these research works mainly focus on the composites of rGO with chemical synthesis polymers. The compounds of CVD-G with electrochemical synthesis polymers are still rarely studied. Recently, we developed two hybrid sensors in which an ultrathin PPy layer was synthesized on CVD-G and rGO by electropolymerization [134,135]. Hereafter, they are called as PPy/CVD-G and PPy/rGO sensor, respectively. The electropolymerization takes advantage of the high conductivity of graphene as an electrode for the PPy deposition. The unique characteristics of graphene simultaneously allows for detecting a modification in resistance and a reduction in the PPy thickness down to the nanoscale range, then reducing the diffusion time.

In Figure 11a, the resistance response of a typical PPy/rGO sensor for 1 ppm NH_3_ is presented at room temperature. A repeatable response of 12.6 ± 0.8% is obtained after 10 injections. Figure 11b gives the sensor responses to NH_3_ concentrations from 1 to 4 ppm, with a sensitivity of 6.1%/ppm. The key parameters of two sensors are added in Table 1 for easy comparison. Two sensors display specific sensing to NH_3_, high sensitivity, quick and reversible response, good stability, and immunity to humidity. These performances are associated with the synergistic effect of the ultrathin PPy layer and graphene.

The sensing mechanism of the PPy/CVD-G sensor for NH_3_ is described as follows. The PPy layer behaves like a p-type semiconductor [148]. When it is exposed to NH_3_, its resistance increases due to the fact that the donor electrons of NH_3_ molecules deplete the holes in the *p*-type PPy. The donor electrons can also transfer to *p*-type CVD-G film through the ultrathin PPy layer, in certain way thereby increasing the *p*-type CVD-G film resistance. On the other hand, the *p*-type CVD-G film is not only the growth support of the PPy layer, but also the conductive pathway of the electron transfer. The high conduction of the CVD-G film accelerates the sensor response and recovery. Additionally, the porous nature of the ultrathin PPy layer makes a major contribution to the sensor selectivity, response speed, and immunity to humidity. The synergistic effect of both materials promotes the sensor performance.

The PPy/rGO sensor and the PPy/CVD-G sensor have the same sensing mechanism when exposed to NH_3_. Nonetheless, the PPy synthesis or growth is more effective on the rGO film than on the CVD-G film. The explanation is that the hydrophilic oxygen groups (hydroxyl and epoxy groups) in the rGO provide more nucleation sites for the PPy layer during the electropolymerization. The PPy/rGO sensor, thus, has a higher availability of the surface activation for NH_3_. Moreover, the formation of π–π stacking between rGO and PPy results in a faster transport of electrons [149]. These facts explain why the PPy/rGO sensor presents a higher sensitivity and quicker response compared to the PPy/CVD-G sensor.

In summary, non-functionalized graphene is chemically inert and nonselective to NH_3_ at room temperature. Therefore, the study on the functionalization materials is critical to fabricate high performance graphene sensors for NH_3_ detection. The synergistic effect of graphene and functional materials significantly improves the sensor performance. As discussed above, there is no unified sensing mechanism for the graphene NH_3_ sensors functionalized by different materials. However, all of the sensor responses depend on their resistance change caused by electrons transfer between NH_3_ and the related sensing layers. High conductance of graphene provides an efficient pathway for electrons transfer, thereby accelerating response and recovery of the sensor. Each functional material plays a key role in the sensitivity and selectivity of the sensors, as listed in Table 2. The most widely used functionalization methods are also listed in this table. Graphene sensors decorated with metallic nanoparticles (MNPs) have achieved a detection limit down to ppb level, but noble MNPs are not compatible with CMOS technology. The mixtures of graphene with metal oxides may improve the selectivity and decrease the response/recovery times. However, most of the mixture materials still need elevated operating temperatures. Organic molecules are the other option for graphene functionalization. Most of them have a short lifetime and are toxic. The graphene functionalized by polymers has a long lifetime, low cost, operation at room temperature, and compatibility with CMOS technology. As a consequence, conducting polymers, particularly, ultrathin ones, are ideal materials for graphene functionalization.

## 5. Challenges and Optimization Pathways

Although a lot of graphene-based NH_3_ sensors have been studied and developed, great challenges still exist for their practical applications. In this section, we will analyze the hindering factors and propose related solutions.

### 5.1. Requirement of Graphene Quality for Sensor Applications

Hwang et al. [150] studied the response of mono-, bi-, and tri-layered graphene sensors to NH_3_. They found out that the number of layers of graphene has no obvious influence on the sensor’s sensitivity. On the contrary, Crowther et al. [151] indicated that gas molecules were mainly adsorbed on the top layer, and the sensitivity of graphene gas sensors decreases when the number of layers increase. The simulation results of Song et al. [152] explain the reason that the charge ratio of integral electrons in total density of states for monolayer graphene is three times more than that of triple layer graphene. Recently, Samaddar et al. [153] in a review article described that the interlayer space in GO structures acts as a barrier for the adsorption of NH_3_ molecules. Therefore, the multiple-layer graphene has poor sensing performance.

Our research results reveal that the PPy/rGO sensor is superior to the PPy/CVD-G in terms of performance. In essence, the PPy/rGO sensor can be produced more easily and at a cheaper cost. However, the PPy/rGO sensor reproducibility is still a key issue. Unlike CVD-G, the size, thickness, and uniformity of the GO dispersions are not well controlled presently. As shown in Figure 12, some clusters are formed by the accumulation of multiple-layer rGO [154]. In this case, NH_3_ molecules adsorbed on the cluster top cannot influence electric current passing through the cluster bottom. This means that the resistance modulation is not very effective, degrading the sensor performance.

Our results suggest that the unique application of the NH_3_ sensors does not absolutely need high-quality single-layer graphene, such as the CVD-G. However, for the integration of the NH_3_ sensors with graphene electron devices in CMOS circuits, CVD-G is a good candidate. rGO is an abundant material and it is very easy to be prepared. However, the sensitivity and response time obviously degrade as the rGO thickness increases. For mass production and commercial application, we have to find appropriate methods to well control the thickness of the rGO for improving the sensor reproducibility. Recently, new approaches for large-scale production of high-quality graphene are being developed and studied, such as industrially viable water-phase exfoliation method [155] and non-electrified electrochemical exfoliation method [156].

### 5.2. Enhancement of Specific Surface Area

The functionalization is the core of the graphene sensor development. It provides specific links and binding sites. The specific links act as bridges to enhance the temporary connection between NH_3_ molecules and graphene. During functionalization, the specific surface area of pristine graphene should be kept unchanged or even increased for obtaining high sensitivity. Tamersit et al. [157] established a theoretical model for sensing gases, in which a double-gate graphene FET was proposed to increase the exposed sensing surface. In their sensor, the pristine graphene channel was surrounded by a dielectric layer. The conducting polymer or catalytic metal were suggested as gate materials (sensing elements), which were attached to top and bottom surfaces of the dielectric layer. In this case, a high sensitivity can be achieved when the double-gate FET operates in the subthreshold region. Cadore et al. [158] also found out that the graphene sensor with both exposed surfaces (suspending) is more sensitive than that with single exposed surface (on substrate). This is contributed to the enhancement of specific surface area of the graphene channel, namely, the increase in reaction sites to NH_3_ by a factor of 2.

Usually, the specific surface area of thick GO dispersions is degraded. Teradal et al. [159] proposed to use the porous GO scaffold for increasing the specific surface area. In their approach, the porous GO scaffold was functionalized by phenyl, dodecyl, or ethanol depending on different target gases. For NH_3_ sensing, phenyl was used. Designing appropriate structures (such as nanomesh or nanoribbon) can also increase the specific surface area of graphene or the edges of graphene. Based on the CVD-G film, Paul et al. [160] constructed a nanomesh graphene sensor by combining polystyrene nanosphere lithography and reactive ion etching. The sensor exhibited a sensitivity of 0.71%/ppm towards NH_3_ with a limit detection of 160 ppb at room temperature. This value is significantly better than its film counterpart due to more edges of the nanomesh graphene. Three-dimensional (3D) structures may be another pathway to increase the specific surface area of graphene [161,162].

### 5.3. Other Treatment Methods of Pristine Graphene

Besides the functionalization methods discussed above, the fluorination or fluorine doping is the other functionalization technology to increase the sensitivity and selectivity of the graphene sensors. Zhang et al. [163] used a plasma-fluorinated monolayer CVD-G to fabricate NH_3_ sensor, which exhibited fast response (30 s) and high sensitivity (3.8%/ppm) at room temperature. The fluorine atoms interact with the graphene matrix to form covalent C−F bonds, which enhance physisorption of NH_3_ molecules, thereby improving the sensor performance. Kim et al. [164] exploited a NH_3_ sensor based on chemically fluorinated GO flakes. The sensor achieved the theoretical detection limit of ~6 ppb at room temperature. The other trends are to synthesis novel multicomponent materials and hybrid nanostructures for improving sensor performance. Su at al. [165] proposed graphene-based ternary composites for NH_3_ sensing, such as the ternary nanocomposite film of Pd-NPs/SnO_2_/rGO. Recently, they reported the results of the ternary film (Pd-NPs/TiO_2_/rGO) for NH_3_ sensing [166]. This sensor strongly responded to low concentrations of NH_3_ at room temperature. For 50 ppm NH_3_, the response of the PPy/TiO_2_/graphene nanoparticles sensor was 102.2% [167], while the responses of the PPy/ZnO sensor and the PANI/rGO sensor were about 50% [18,126]. The ternary-composite sensor has much higher sensitivity than the two-composite sensors.

### 5.4. Substrate Engineering

The sensor substrate also plays an important role for the interaction of the sensing layer with target gas. Cadore et al. [158] compared the performance of the mechanical exfoliation graphene FET sensors on different substrates, such as SiO_2_, talc (Mg_3_Si_4_O_10_(OH)_2_), and hexagonal boron nitride (hBN). They found out that for the NH_3_ detection the FET sensor on G/hBN-substrate exhibited higher response and faster recovery than the FET sensor on G/SiO_2_-substrate. The electron transfer from NH_3_ molecules to the graphene sensing layer is strongly dependent on the distance between the sensing layer and substrate. Aziza et al. [168] compared the performance of the CVD-G resistive sensors on mica (AC_2-3_T_4_O_10_X_2_) and SiO_2_ substrates. The sensitivity of the sensor on G/mica-substrate is higher than that of the sensor on G/SiO_2_-substrate, because the mica substrate induces more *p*-doping in graphene. Therefore, the substrate engineering should be put under consideration for the further improvement of the sensor performance, particularly, in the choice of the flexible substrates.

### 5.5. Mass Production of Graphene NH_3_ Sensors

Up to now, the graphene NH_3_ sensors are still in the laboratory. To move them from the laboratory level to commercial and practical applications, the following challenges exist.

Mass production of graphene NH_3_ sensors first needs large-area few-layer graphene with controlled structural quality. CVD-G is the most appropriate choice. However, high growth temperature (above 1000 °C), transferring from metal foils to target substrates, and catalytic metal foil waste make the CVD process complex, expensive, and have a negative impact on the environment. By contrast, rGO can be produced in large quantities and converted from GO by thermal treatment or chemical reduction. Particularly, rGO can be deposited by simple methods such as drop coating, spin coating, and inkjet printing. Although the rGO synthesis is quite easy and cheap, the rGO properties (including size, thickness, and quality) are difficult to be well controlled. In addition, high contact resistance between rGO and metal electrodes degrades the sensor sensitivity.

Deokar et al. [169] synthesized wafer-scale few-layer graphene on Cu/Ni films for NH_3_ sensing applications. A detection limit of 1 ppm and response time of 30 min were observed for a typical sensor at room temperature. However, the characterization of the batch sensors has not been reported yet. Our PPy/G and PPy/rGO sensors were fabricated on 3-inch wafers. However, the electropolymerization was a post process step, which was carried out one sensor by one sensor at a time. To move the present process from laboratory concept to industrial application, it is necessary to functionalize all the sensors on one wafer once. In reality, the technique to produce the graphene NH_3_ sensors with a high yield was not been found due to reproducibility issues. In addition, for the feasibility of a sensing system, uniformity across the batch is a more relevant parameter than the sensor itself [170,171]. Furthermore, long-term stability is another key feature for practical applications of the sensors. In order to overcome these problems, one should further develop more uniform, repeatable, and stable graphene functionalization methods and sensor fabrication techniques that will reduce the sensor-to-sensor variation. The sensor operation in real-life conditions (room temperature, 50–55% RH, and atmospheric pressure) is also very crucial.

From what has been discussed above, although CVD-G presents high quality, it requires high growth temperature and catalytic metal foils. The rGO is an easily synthesized, cheap, and defective graphene. However, the properties of the GO dispersions should be well controlled. The suspended graphene and designed structures can increase the specific surface area or the edges of graphene. New and innovative functionalization technologies, such as fluorination and graphene-based ternary composites, are other ways for improving sensing properties. In addition, the substrate choice is also important for the further improvement of the sensor performance. For mass production and practical applications, the sensor reproducibility, uniformity, stability, and immunity to humidity should be further exploited and investigated.

## 6. Possible Opportunities and Future Directions

Graphene NH_3_ sensors can be assembled with commercially available electronic devices and modules, as well as previously fabricated chips to build a prototype. With application software, the prototype would wirelessly transfer the measurement data to a mobile phone or computer. Thus, one could continuously and remotely monitor the NH_3_ concentration in a real-life environment.

Our research indicates that the combination of graphene and conducting polymers is very promising as a chemical sensing material. As long as the electropolymerization could be carried out on wafer scale, it could be possible to integrate the conducting polymer/graphene sensors with CMOS circuits onto the same chip for monitoring different gases. The electronic nose consists of a gas-sensor array to detect various gases through multiple sensors. Graphene could be used as the sensing layers in the array. Then, the sensing layer of each sensor is functionalized by different materials or methods according to the requirements of target gases. This could be of great interest for fabricating the electronic nose fully integrated with CMOS circuits for accurately detecting different gases and biochemical and chemical molecules. It also could be of great interest for fabricating lab-on-chip in the field of environmental monitoring. Due to the current boom of IoT and connected devices, there is a growing demand for low-power devices. Graphene-based gas sensors with good performance are more advantageous in this topic than gas sensors based on heated metal oxides in the market, since the former requires less energy than the latter.

## 7. Conclusions

Graphene is a potential material for ammonia (NH_3_) sensing due to its large specific surface area and high carrier mobility. However, the non-functionalized graphene is nonselective and weakly binding with NH_3_ molecules at room temperature. Fortunately, it can be easily modified by functional groups. Functionalized graphene will occupy an important position in the future of gas-sensing materials. This paper summarizes the graphene sensors functionalized by different materials for NH_3_ detection, including metallic nanoparticles, metal oxides, organic molecules, and conducting polymers. Among them, the graphene functionalized by conducting polymers has a long lifetime, low cost, operation at room temperature, and compatibility with CMOS technology. Recently, we functionalized graphene and reduced graphene oxide by ultrathin polypyrrole (PPy) through electropolymerization. The synergistic effect of graphene and the ultrathin PPy significantly improves the sensor performance. Graphene increases the conductivity of the polymer film, accelerating the sensor response and recovery. On the other hand, the polymer film provides the specific reaction sites for NH_3_ adsorption, defining the sensor selectivity and enhancing the sensor sensibility. Overall, the sensor performance is strongly influenced by the charge transfer mechanisms, the functional molecules, and graphene itself. The functionalized graphene NH_3_ sensors will have a bright perspective due to the advantages of high sensitivity, fast response, great selectivity, low power consumption, cost-effectiveness, and operation in real-life condition. They are expected to be applied in the fields of lab-on-chip, electronic nose, Internet of Things for industrial control, agricultural storage, environmental monitoring, disease diagnosis, wearable health, ecological protection, and public safety.

## Figures and Tables

**Figure 1 sensors-21-01443-f001:**
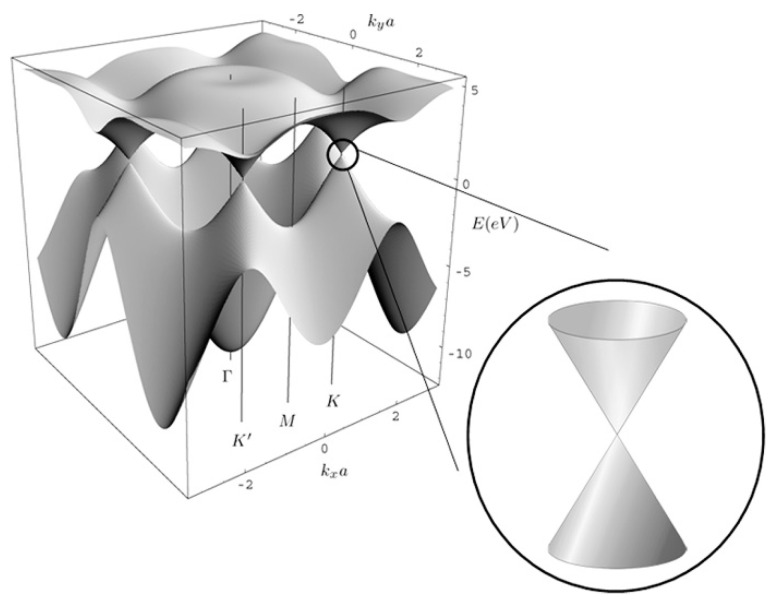
Schematic diagram for band structure of single-layer graphene. The inset on the lower right highlights the linear dispersion at the Dirac point. Reproduced from (Ernie W. Hill, [31]), Copyright 2011, IEEE.

**Figure 2 sensors-21-01443-f002:**
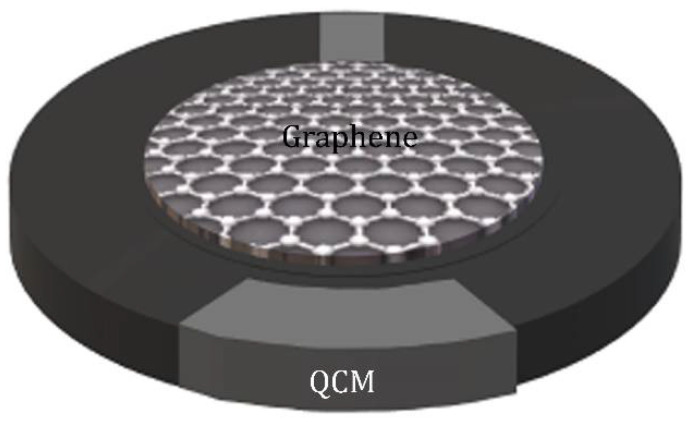
Schematic diagram of a graphene-based quartz crystal microbalance sensor.

**Figure 3 sensors-21-01443-f003:**
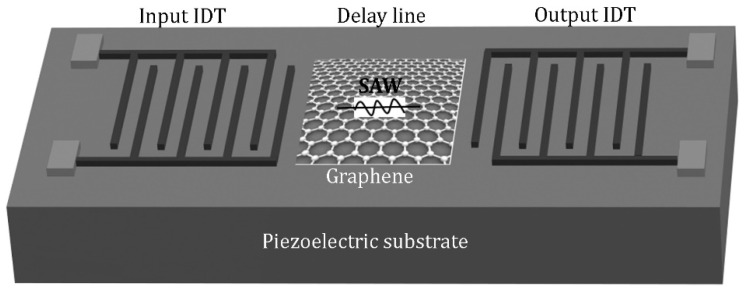
Schematic diagram for a graphene-based surface acoustic wave sensor.

**Figure 4 sensors-21-01443-f004:**
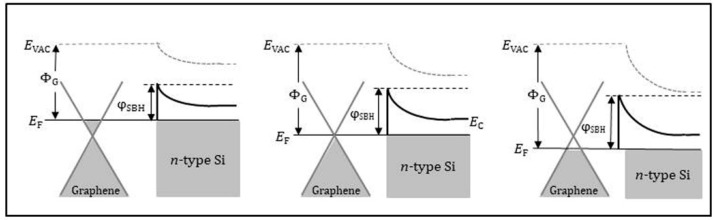
Schematic energy band diagram of the graphene/*n*-type Si interface for the pristine graphene state (**middle**), the graphene exposed to electron-donor gas (**left**) and the graphene exposed to electron-acceptor gas (**right**). *E*_VAC_, *E*_C_, Φ_G_, *E*_F_, and *ψ*_SBH_ indicate the vacuum energy, conduction band, graphene work function, Fermi level, and Schottky barrier height, respectively. Reproduced from (Hye-Young Kim, [55]), Copyright 2013, American Chemical Society.

**Figure 5 sensors-21-01443-f005:**
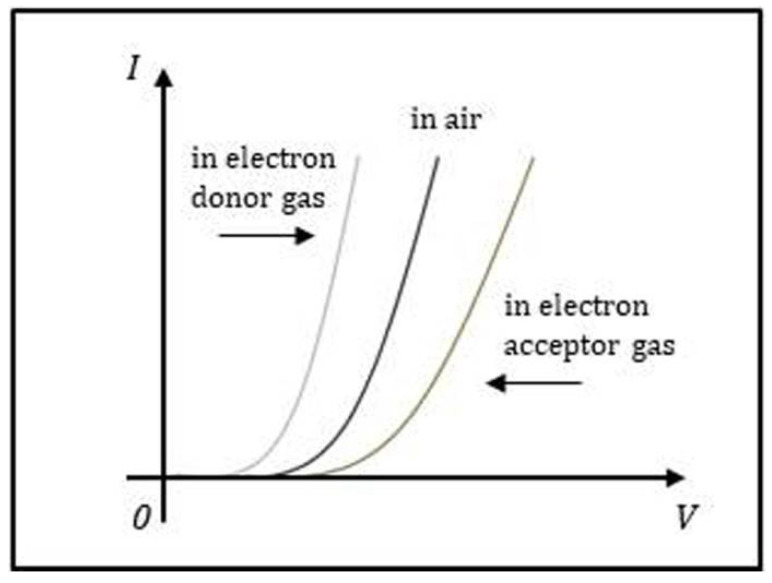
Schematic illustration of the *I*–*V* characteristics for a graphene/*n*-type Si Schottky diode sensor in air, exposed to electron-donor gas and electron-acceptor gas.

**Figure 6 sensors-21-01443-f006:**
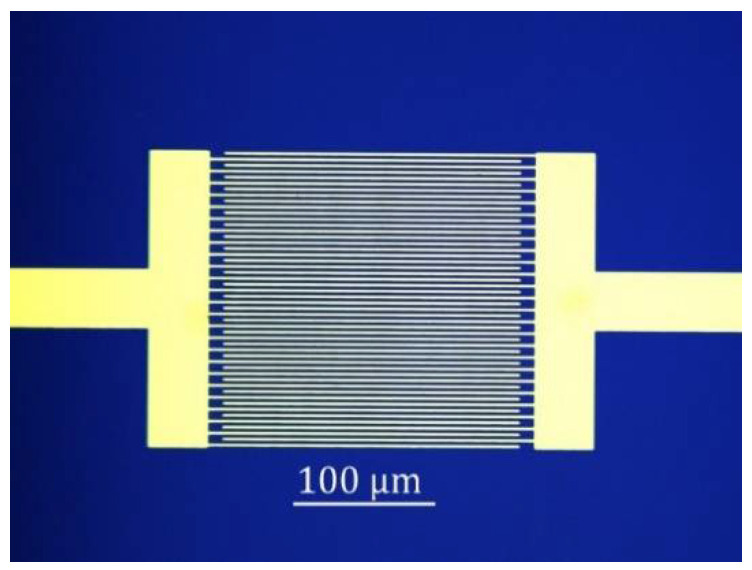
Optical image of Au interdigitated electrode (IDEs) on SiO_2_/Si substrate for a graphene chemiresistor sensor. The sensing area is of 200 × 200 µm^2^. The finger width and finger space are consistent (2 µm).

**Figure 7 sensors-21-01443-f007:**
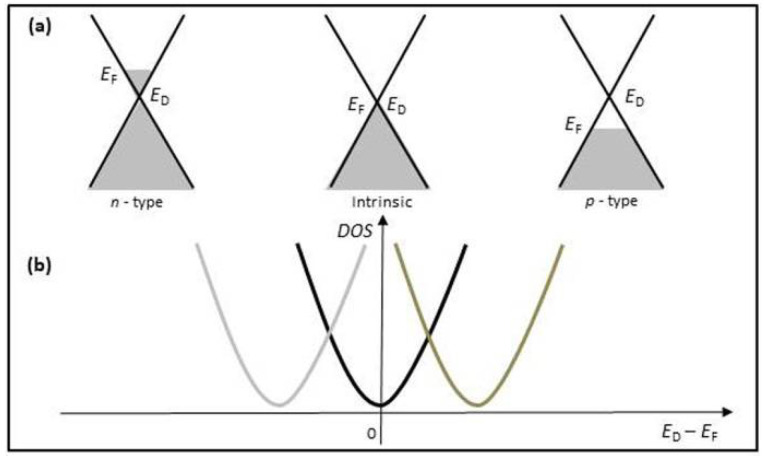
(**a**) Energy band diagrams of graphene: pristine graphene (**middle**), *n*-type graphene (**left**), and *p*-type graphene (**right**). (**b**) Graphene resistance as a function of gate voltage. *E*_F_ and *E*_D_ indicate Fermi level and Dirac point, respectively.

**Figure 8 sensors-21-01443-f008:**
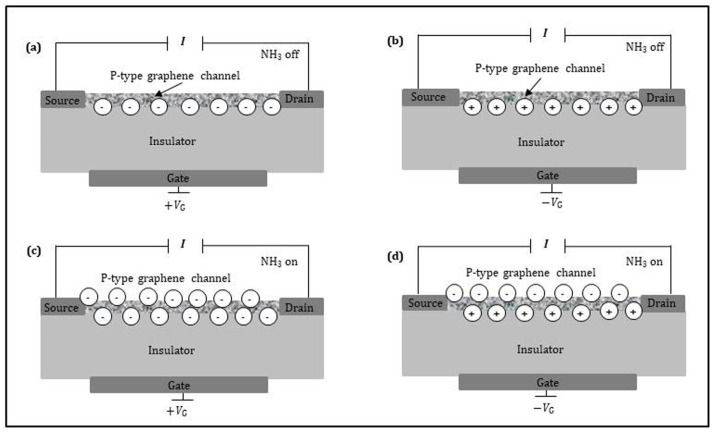
Schematic cross-sections of a graphene FET sensor applied by: (**a**) positive gate voltage for NH_3_ off; (**b**) negative gate voltage for NH_3_ off; (**c**) positive gate voltage for NH_3_ on; (**d**) negative gate voltage for NH_3_ on.

**Figure 9 sensors-21-01443-f009:**
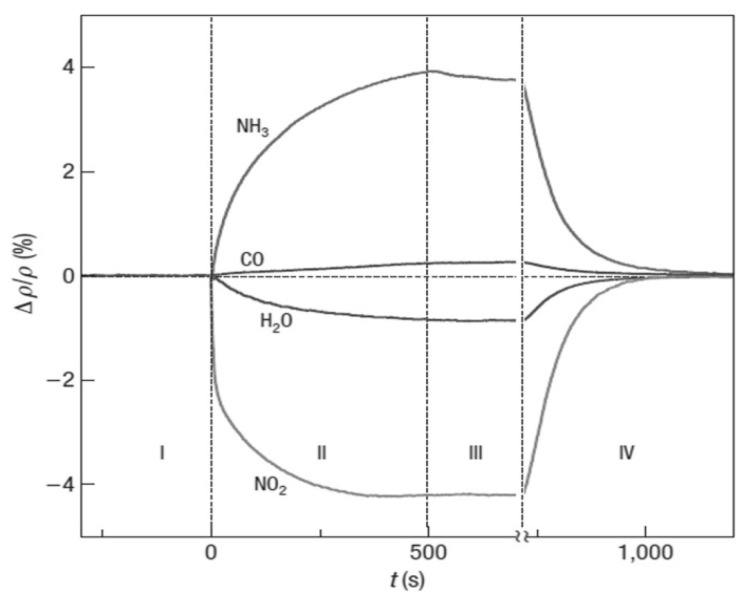
Resistance responses of a pristine graphene sensor to 1 ppm ammonia (NH_3_), carbon monoxide (CO), nitric oxide (NO), and water vapor (H_2_O) in pure helium or nitrogen at atmospheric pressure, at 150 °C. Reproduced from (F. SCHEDIN, [23]), Copyright 2007, Nature**** Publishing Group.

**Figure 10 sensors-21-01443-f010:**
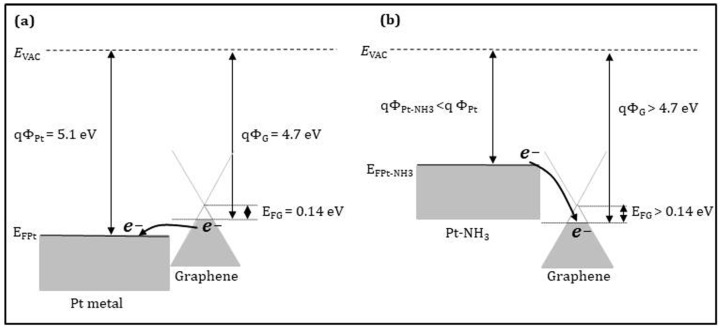
Energy band diagrams (**a**) for Pt and *p*-type graphene and (**b**) for Pt-NH_3_ phase and *p*-type graphene. *E*_VAC_, Φ_Pt_, *E*_FPt_, Φ_G_, *E*_FG_, Φ_Pt-NH3_ indicate the vacuum energy, Pt work function, Pt Fermi level, graphene work function, graphene Fermi level, and Pt-NH_3_ work function, respectively.

**Figure 11 sensors-21-01443-f011:**
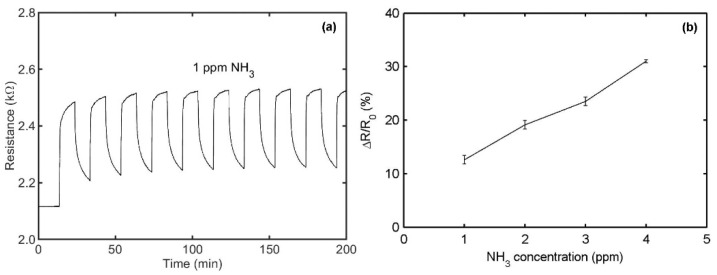
Sensing behaviors of a typical PPy/reduced graphene oxide (rGO) sensor at relative humidity of 50% and at 20 °C: (**a**) The sensor resistance behavior to 1 ppm NH_3_ for 10 cycles. (**b**) The sensor resistance response as a function of NH_3_ concentration. Reproduced from (Xiaohui Tang, [135]), Copyright 2020, Elsevier.

**Figure 12 sensors-21-01443-f012:**
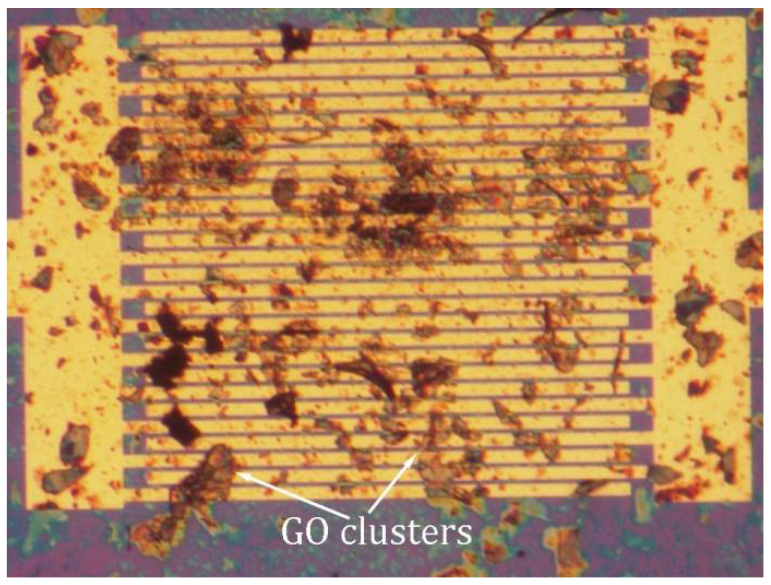
Optical image of graphene oxide (GO), dispersions (in water) on interdigitated electrodes over SiO_2_/Si substrate. The GO dispersions (deposited by ultrasonic spray pyrolysis) are agglomerated to form clusters. Reproduced from (Xiaohui Tang, [135]), Copyright 2020, Elsevier.

**Table 1 sensors-21-01443-t001:** Graphene sensors functionalized by conductive polymers for NH_3_ detection at room temperature.

Sensor Type	Sensing Material	Sensitivity	Detection Limit	Response Time	Recovery Time	Ref.
Chemiresistor	PPy/CVD-G	0.9%/0.1 ppb	0.1 ppb	10 s		[136]
Chemiresistor	Py/rGO	2.4%/ppb	1 ppb	1.4 s + IR	76 s + IR	[133]
Chemiresistor	PANI/rGO	3.65%/20 ppm	1 ppm	50 s	23 s	[137]
Chemiresistor	PEDOT:PSS/G	9.6%/500 ppm	25 ppm	180 s	300 s	[138]
Chemiresistor	PPy/rGO	34.7%/500 ppm	3 ppm	400 s		[131]
Chemiresistor	PANI/rGO	59.2%/50 ppm	50 ppm		4 min	[126]
Chemiresistor	PANI/rGO	14%/ppm	0.2 ppm			[139]
Chemiresistor	PPy/rGO	4.5%/ppm	0.1 ppm	118 s	122 s	[140]
Chemiresistor	PANI/rGO	37.1%/50 ppm	20 ppm	18 min	2 min	[141]
Chemiresistor	PANI/rGO	344%/100 ppm	10 ppm	20 s	27 s	[142]
Chemiresistor	PPy/rGO	22%/100 ppm	5 ppb	134 s + IR	310 s + IR	[132]
3D structure	PPy/3D-rGO	3%/ppm	330 ppb	5 s	20 s	[143]
Chemiresistor	ANI/rGO	10.7%/5 ppm		18 min	3 min	[144]
Schottky diode	PEDOT:PSS/rGO	9.7%/ppm	1 ppm	95 s	121 s	[145]
Chemiresistor	PPy/rGO		1 ppm			[146]
SPR	PMMA/rGO		10 ppm	60 s		[147]
Chemiresistor	PPy/CVD-G	1.7%/ppm	1 ppm	2 min	5 min	[134]
Chemiresistor	PPy/rGO	6.1%/ppm	<1 ppm	1 min	5 min	[135]

rGO: reduced graphene oxide, G: graphene, PPD: p-phenyldiamine, DMMP: dimethyl methyl phosphonate, PANI: polyaniline, ANI: aniline, Py: pyrrole, COP: chemical oxidative polymerization, PEDOT: PSS poly (3,4-ethylenedioxythiophene): poly (styrene sulfonate), IR: infrared light, SPR: surface plasmon resonance.

**Table 2 sensors-21-01443-t002:** Main sensing mechanisms of functionalized graphene NH_3_ sensors and graphene functionalization methods based on metallic nanoparticles, metal oxides, organic molecules, and conducting polymers.

Functional Materials	Sensing Mechanisms	Functionalization Methods
Metallic nanoparticles: Au Ag Pt Pd	Sensor response depends on electrons transfer from NH_3_ to G or rGO. Metallic nanoparticle acts as a catalyst to increase the reaction between NH_3_ and G or rGO.	Electrochemical method Hydrothermal reduction Physical vapor deposition Layer-by-layer self-assembly Electrostatic interactions Galvanic replacement reaction
Metal oxides: SnO_2_ V_2_O_2_ ZnO TiO_2_ Cu_2_O WO_3_ In_2_O_3_	Sensor response depends on electrons transfer from NH_3_ to C-O-M- bonds. Metal oxide acts as a predominating NH_3_ receptor, and G or rGO accelerates sensor response and recovery.	Thermal reduction Hydrothermal reduction Precipitation Electrospinning One-pot polyol Pulse laser deposition
Organic molecules: Rose Bengal (RB) nanocomposite	Sensor response depends on electrons transfer from NH_3_ to G or rGO. Functional groups in RB act as extra active sites and facilitate more binding of NH_3_.	Drop casting π coupling
Bromophenol blue (BPB) Co-porphyrin	Sensor response depends on electrons transfer from NH_3_ to G or rGO. Protonated acidic rings and electrophilic protons in BPB act as NH_3_ attractors (electrons transferring between BPB and G or rGO via coupling π bonds).	Layer-to-layer stacking π coupling Spin coating
Polymers: Polypyrrole Polyaniline Others (PTh, PBuA, and PVDF)	Sensor response depends on electrons transfer from NH_3_ to polymers via its conjugated bonds. G or rGO provides an efficient pathway for electron transfer, accelerating sensor response and recovery.	Electropolymerization Electrochemical Chemical Pyrrole reaction

G: graphene, rGO: reduced graphene oxide, M: metal, PTh: polythiophene, PBuA: poly (butyl acrylate), PVDF: poly (vinylidene fluoride).

## Data Availability

Not applicable.

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
