# Peer review of "A Review on Functionalized Graphene Sensors for Detection of Ammonia"

_sensors, 2021, doi:10.3390/s21041443_

Round 1

Reviewer 1 Report

Dear Authors,

the paper is well organised and pleasant to read. The pertinent literature is cited too. I would just suggest to correct some typos:

  • Line 33 & 564: "NH3", "3" should be subscript,
  • Line 36: "valve" should be "value",
  • Line 53: not clear: how can the sensor detect NO2 if in vacuum? Check, the original paper, please,
  • Line 63: "cleanly" should be "clearly",
  • Lines 194 & 198: "absorb" should be changed in "adsorb",
  • Line 212: "...nucleophile, can be attracted" should be "...nucleophile, it can be attracted",
  • Line 257: "thin silicon dioxide (SiO2)" should be "thin silicon dioxide (SiO2) layer",
  • Line 373: "of is weak graphene" should be "of graphene is weak",
  • Line 413: "showed" and not "shown",
  • Line 479: "bromophenol" and not "bromphenol ",
  • Line 506: "which can be act as a gas" should be "which can act as a gas",
  • Table 1: what is IDAs? Why did you have separated IDE and chemiresistive sensors?
  • Line 556: "0.8%" without any blank space,
  • Line 557: "of 6.1%/ppm" as well,
  • Line 586: "chemically" and not "chemical",
  • Line 704: "produce" and not "product".

Best regards.

Reviewer

Author Response

Reviewers' comments:

Reviewer #1: the paper is well organized and pleasant to read. The pertinent literature is cited too. I would just suggest to correct some typos:

Many thanks to the reviewer! We have very much appreciated the reviewer’s comments and made all corrections accordingly. Please see the red parts in the revised version. We would like to note: according to the second reviewer’s comments, we rewrote some paragraphs so the line numbering has changed in the revised version.

  • Line 33 & 564: "NH3", "3" should be subscript,

We have changed all notations in the revised version.

  • Line 36: "valve" should be "value",

       Revised in Line 36

  • Line 53: not clear: how can the sensor detect NO2 if in vacuum? Check, the original paper, please,

Thanks for the review! We have made a mistake. The correct sentence was changed in Line 54.

  • Line 63: "cleanly" should be "clearly",

Revised in Line 64

  • Lines 194 & 198: "absorb" should be changed in "adsorb",

We have changed all instances in the revised version.

  • Line 212: "…nucleophile, can be attracted" should be "nucleophile, it can be attracted",

This sentence has been rewritten in Line 215.

  • Line 257: "thin silicon dioxide (SiO2)" should be "thin silicon dioxide (SiO2) layer",
  • It has been changed in Line 263.
  • Line 373: "of is weak graphene" should be "of graphene is weak",

It has been changed in Line 386.

  • Line 473: "showed" and not "shown",

We have changed all instances in the revised version.

  • Line 479: "bromophenol" and not "bromphenol ",

Revised in Line 506

  • Line 506: "which can be act as a gas" should be "which can act as a gas",

Revised in Line 534

  • Table 1: what is IDAs? Why did you have separated IDE and chemiresistive sensors?

In the previous version, we tried to separate the chemiresistive sensors into 2 groups: one group has two electrode pads and the other has interdigitated electrodes (IDE). To avoid the confusion, we will call them as chemiresistive sensors in Table 1 in the revised version. Thanks for your question!

  • Line 556: "0.8%" without any blank space,

Revised in Line 574

  • Line 557: "of 6.1%/ppm" as well,

Revised in Line 575

  • Line 586: "chemically" and not "chemical",

Revised in Line 604

  • Line 704: "produce" and not "product".

Revised in Line 740

Best regards.

The authors

Reviewer 2 Report

Overall, this review study has represented valuable information regarding the functionalized graphene sensor for detection of Ammonia. However, this paper has some shortcomings in the direction of functionalization mechanism. Also need to provide some better citation which will help the readers to get better and accurate information regarding the detection of ammonia by utilizing the functionalized graphene sensors.

There are some points need to be clarified:

1. Authors need to focus on the abstract and the results. Here, in the abstract authors have mentioned “The various sensing mechanism of functionalized graphene gas sensors are explained and compared”. However, in the paper authors have only compared the graphene functionalized by conducting polymers. Therefore, the authors need to provide the comparison table by considering Functionalization of graphene using metallic nanoparticles, metal oxides, organic molecules, and compare with conducting polymers.

2. Authors need to clearly explain the functionalization method of graphene (Metallic nanoparticles, Metal oxides, Organic molecules) and the mechanism behind the response to NH 3 gas detection.

3. In this review most of the gas sensor use functionalized RGO as the sensing material. Authors need to provide some more discussion regarding the functionalized CVD growth graphene for ammonia gas sensor.

4. Authors need to clearly compare and discuss the sensor based on pristine graphene and functionalized graphene.

5. Line 33: detecting NH3 concentration 3 should be subscripted.

6. Line 47: the font size of reference [18,19,20] should be same with other references.

7. Line 188-193: The part about Mass sensitive layer could be rearranged to give a better explanation. The definition is too general, the working principle is not quite clear due to the sentence arrangement/punctuation.   8. On the FET part, the authors suggest the measurement technique should be improved. The possibility of the improvements that have been done in other cases might be good to be suggested.   9. The part of Metal oxide is very few discussed since it is referred to another review that has done by the authors. However, it might be better if more discussion about example case of improvement on the sensing performance could be added.     10. On the part of the Polymers functionalization, the authors mention the excellent performance of polypyrrole and polyaniline. It could be better if the claimed performance is cited from reference(s).   11. Not all terms in Table 1 are explained (IDAs).

Author Response

Reviewers' comments:

Reviewer #2: Overall, this review study has represented valuable information regarding the functionalized graphene sensor for detection of Ammonia. However, this paper has some shortcomings in the direction of functionalization mechanism. Also need to provide some better citation which will help the readers to get better and accurate information regarding the detection of ammonia by utilizing the functionalized graphene sensors.

We would like to thank the reviewer’s helpful comments on this manuscript! We have made a major correction in the revised version.

There are some points need to be clarified:

  1. Authors need to focus on the abstract and the results. Here, in the abstract authors have mentioned “The various sensing mechanism of functionalized graphene gas sensors are explained and compared”. However, in the paper authors have only compared the graphene functionalized by conducting polymers. Therefore, the authors need to provide the comparison table by considering Functionalization of graphene using metallic nanoparticles, metal oxides, organic molecules, and compare with conducting polymers.

Following the reviewer’s suggestion, we added a comparison table in the revised version (please see Table 2 on page 17). Moreover, we have also rewritten the last paragraph in the end of Section 3 to explain and compare the various sensing mechanisms of functionalized graphene NH3 sensors. Please see the red part on page 16.

  1. Authors need to clearly explain the functionalization method of graphene (Metallic nanoparticles, Metal oxides, Organic molecules) and the mechanism behind the response to NH 3 gas detection.

We summarized main graphene functionalization methods by metallic nanoparticles, metal oxides, organic molecules, and conducting polymers in Table 2 on page 17. As we can see in the table, a variety of functionalization methods exist in the literature. Therefore, we could not describe them all in detail due to the length limitation of the article. Moreover, this is beyond the goal of this article. Please see Line 68.

  1. In this review most of the gas sensor use functionalized RGO as the sensing material. Authors need to provide some more discussion regarding the functionalized CVD growth graphene for ammonia gas sensor.

We fully agree with the reviewer. Most of our discussion focused on the functionalized rGO NH3 sensors in the previous version, since the functionalized CVD-G NH3 sensors were rarely studied in the literature. Particularly, the conducting polymer/CVD-G NH3 sensors were even more infrequent. In the revised version, we provided the discussion and explanation for two NH3 sensors: SnO2/CVD-G and V2O5/CVD-G in section 3.2 (Metal oxides) on page 12. Please also see our answer to your question 9. Moreover, we cite the following references on the functionalized CVD-G NH3 sensors in the revised version.

(1) Gautam, M.; Jayatissa, A.H. Ammonia Gas Sensing Behavior of Graphene Surface Decorated with Gold Nanoparticles. Solid-State Electronics 2012, 78, 159–165, doi:10.1016/j.sse.2012.05.059.

(2) Song, H.; Li, X.; Cui, P.; Guo, S.; Liu, W.; Wang, X. Morphology Optimization of CVD Graphene Decorated with Ag Nanoparticles as Ammonia Sensor. Sensors and Actuators B: Chemical 2017, 244, 124–130, doi:10.1016/j.snb.2016.12.133.

(3) Hijazi, M.; Stambouli, V.; Rieu, M.; Tournier, G.; Pijolat, C.; Viricelle, J.-P. Sensitive and Selective Ammonia Gas Sensor Based on Molecularly Modified SnO2. Proceedings 2017, 1, 399, doi:10.3390/proceedings1040399.

(4) Kodu, M.; Berholts, A.; Kahro, T.; Eriksson, J.; Yakimova, R.; Avarmaa, T.; Renge, I.; Alles, H.; Jaaniso, R. Graphene-Based Ammonia Sensors Functionalised with Sub-Monolayer V2O5: A Comparative Study of Chemical Vapour Deposited and Epitaxial Graphene †. Sensors 2019, 19, 951, doi:10.3390/s19040951.

  1. Authors need to clearly compare and discuss the sensor based on pristine graphene and functionalized graphene.

In the beginning of Section 3 in the revised version, we added a new paragraph to compare and discuss the performance of the pristine graphene and functionalized graphene sensors. Please see Line 362.

  1. Line 33: detecting NH3 concentration 3 should be subscripted.

We have changed all notations in the revised version.

  1. Line 47: the font size of reference [18,19,20] should be same with other references.

Revised on Line 47

  1. Line 188-193: The part about Mass sensitive layer could be rearranged to give a better explanation. The definition is too general, the working principle is not quite clear due to the sentence arrangement/punctuation.

We agree with the reviewer and rewrote this part in page 5 of the revised version.

  1. On the FET part, the authors suggest the measurement technique should be improved. The possibility of the improvements that have been done in other cases might be good to be suggested.

We are preparing a new manuscript, which focuses on the measurement techniques for the FET sensors. Therefore, we prefer to put this part in the coming manuscript. In any case, we added the sentence “For example, an alternating current (AC) is applied to the gate of the graphene FET sensor in order to simultaneously excite the perturbation of target molecular dipoles and modulate the channel charge carriers.  Introducing holes with the gate voltage will favor the adsorption of ammonia while an opposite  voltage will withdraw holes and favor desorption.  ” Please see page 10.

Ref. Kulkarni, G.S.; Zang, W.; Zhong, Z. Nanoelectronic Heterodyne Sensor: A New Electronic Sensing Paradigm. Acc. Chem. Res. 2016, 49, 2578–2586, doi:10.1021/acs.accounts.6b00329.

  1. The part of Metal oxide is very few discussed since it is referred to another review that has done by the authors. However, it might be better if more discussion about example case of improvement on the sensing performance could be added. 

As suggested by the reviewer’s comments, we rewrote Section 3.2 in the revised version, which explains and discusses the sensing mechanisms and functionalization methods for two metal oxide/CVD-G hybrid sensors. Please see page 12.

  1. On the part of the Polymers functionalization, the authors mention the excellent performance of polypyrrole and polyaniline. It could be better if the claimed performance is cited from reference(s).

According to the reviewer’s suggestion, we have cited 5 related references in the revised version (please see Line 513 on page 13).

  1. Not all terms in Table 1 are explained (IDAs).

In our previous version, we tried to separate the chemiresistive sensors into 2 groups: one group has two electrode pads (just 2 electrodes in front of each other) and the other has interdigitated electrodes (IDAs). To avoid any confusion, we will call them chemiresistive sensors in Table 1, in the revised version. Thanks for your question!

Best regards.

The authors
